# Calcium and Potassium Nutrition Increases the Water Use Efficiency in Coffee: A Promising Strategy to Adapt to Climate Change

**Victor Hugo Ramírez-Builes * and Jürgen Küsters**

Center for Plant Nutrition and Environmental Research Hanninghof, Yara International, 48249 Dülmen, Germany; juergen.kuesters@yara.com
* Correspondence: victor.ramirez@yara.com; Tel.: +49-25947980 (ext. 192)

**Abstract:** Coffee (*Coffea* spp.) represents one of the most important sources of income and goods for the agricultural sector in Central America, Colombia, and the Caribbean region. The sustainability of coffee production at the global and regional scale is under threat by climate change, with a major risk of losing near to 50% of today's suitable area for coffee by 2050. Rain-fed coffee production dominates in the region, and under increasing climate variability and climate change impacts, these production areas are under threat due to air temperature increase and changes in rainfall patterns and volumes. Identification, evaluation, and implementation of adaptation strategies for growers to cope with climate variability and change impacts are relevant and high priority. Incremental adaptation strategies, including proper soil and water management, contribute to improved water use efficiency (WUE) and should be the first line of action to adapt the coffee crop to the changing growing conditions. This research's objective was to evaluate at field level over five years the influence of fertilization with calcium ($Ca^{+2}$) and potassium ($K^+$) on WUE in two coffee arabica varieties: cv. Castillo and cv. Caturra. Castillo has resistance against coffee leaf rust (CLR) (*Hemileia vastatrix* Verkeley and Brome), while Caturra is not CLR-resistant. WUE was influenced by yield changes during the years by climate variability due to El Niño–ENSO conditions and CLR incidence. Application of $Ca^{+2}$ and $K^+$ improved the WUE under such variable conditions. The highest WUE values were obtained with an application of 100 kg CaO $ha^{-1}$ $year^{-1}$ and between 180 to 230 kg K2O $ha^{-1}$ $year^{-1}$. The results indicate that adequate nutrition with $Ca^{+2}$ and $K^+$ can improve WUE in the long-term, even underwater deficit conditions and after the substantial incidence. Hence, an optimum application of $Ca^{+2}$ and $K^+$ in rain-fed coffee plantations can be regarded as an effective strategy to adapt to climate variability and climate change.

**Keywords:** climate variability; climate change; WUE; calcium; potassium

## 1. Introduction

Climate change is one of the most significant challenges for humanity. The global mean surface temperature (GMST) is increasing at the rate of 0.2 °C +/−0.1 °C per decade, reaching 1.0 °C above the pre-industrial period (reference period 1850–1900) in 2007 and is projected to reach 1.5 °C above the pre-industrial period between 2030 and 2052, depending on the model and assumptions regarding projected changes to atmospheric greenhouse gases (GHG) levels and climate sensitivity [1]. The most impacted regions worldwide, experiencing increasing GMST, will be those located in the low and mid-latitudes or tropical and extratropical regions [2,3], with projected increases in drought risk, dryness, precipitation deficits, air temperatures increases and heatwaves risk [1].

Climate change is considered one of the biggest challenges for the coffee food chain. The air temperature increases and exacerbates climate variability, affecting the rainfall, solar radiation and air temperature patterns in almost all the coffee regions of the world, including the Americas. Several studies remark that will be exists a reduction in the 50% of

the area suitable for coffee production until the 2050s [4–8]. Those changes in the rainfall and air temperature patterns pose significant challenges to smallholder coffee farmers, where the coffee growth mainly in rain-fed conditions, with limited access to financial and technical support that could help them to respond to changing climate conditions [9,10].

Coffee production (*Coffea* spp.) in Central America, Colombia, and the Caribbean covers 29.65% of the total planted area, with nearly 2,626,949 ha and with gross revenue of USD 2.8 billion [11], providing economic sustainability near 6.6 million, mostly small-holders. Coffee exportations generate wealth and income to the produces countries, apart from its primary social functions. According to Tol [3], developing countries are more vulnerable to climate change for three main reasons: first, a higher share of their economy's activity in agriculture like coffee, second, they tend to be hotter, and third, they tend to have poor adaptive capacity. When we searched for adaptative strategies for the coffee sector, we found several initiatives, for example, migration of coffee areas that are located below 1000 m above sea level (masl), as both species of coffee lose a large share of total sustainability in low altitudes [5]. The migration of coffee areas is not a sustainable adaptation strategy for some farmers and regions. This means that the farmers need to explore and implement other strategies. Some adaptation options may require incremental modifications in current farming practices like tree windbreaks, new crop varieties, on-farm soil and water management; others may need radical changes in production systems structure known as system adaptation and transformation adaptation [12].

Rain-fed crops like coffee are subject to several intermittent stress factors like water stress, heat stress, and nutrient deficit stress, decreasing the yield potential. The $K^+$ depletion, for example, is one of the primary reasons for the degradation of arable land in southeast Asia, Latin America, and the Caribbean [13]. Potassium ($K^+$) and calcium ($Ca^{+2}$) are considered the first and third most demanded nutrients, respectably, by coffee and cherries [14,15]. $K^+$ is a necessary nutrient in coffee to buildup resistance against especially fungal diseases, aside from preventing water stress by regulating turgor pressure, by influencing stomatal opening and closing, directly affect the cherry and bean formation and quality by stimulating enzyme activities, as well carbohydrates synthesis and translocation [16]. In addition, $K^+$ is a critical intracellular agent known to influence osmotic pressures; if the supply of $K^+$ is adequate, cells maintain their turgor within an optimal range, and metabolic processes can occur uninhibited [17].

Waraich et al. [18] described the possible mechanisms to enhance WUE by improving $K^+$ nutrition in crop plants by three main processes (i) maintain the high pH in stoma that reduces photo-oxidative damage and maintain chloroplast membrane; (ii) enhance root growth and decrease the loss of soil moisture that allows the root to explore the soil moisture and maintains turgidity, and (iii) decrease transpiration that increases retention of water in plants and maintains the osmotic potential. According to Grzebisz et al. [19], the uptake of nutrients from the soil solution is governed by two main processes: (i) the transport of ions from the soil solution to the root surfaces and (ii) root growth into the soil patches that are rich in nutrients. The rate of K-ion diffusion towards the root depends on the K-concentration gradient between the root surfaces and the soil solution. As a consequence of the low mobility of the K-ions in soil solution, $K^+$ concentration near roots decreases very quickly. Hence, the application of K fertilizer is important to guarantee an optimum $K^+$ concentration in the soil solution, which in turn ensures a sufficient K uptake by the roots.

The physiological and structural role of $Ca^{+2}$ in alleviating stress conditions has been widely documented [20]. $Ca^{+2}$ plays a vital role in regulating many physiological processes that influence crop growth and response to environmental stress [17]. $Ca^{+2}$ has been recognized as a universal signaling nutrient that allows a proper response of the plants to stress conditions. A recent publication described how $Ca^{+2}$ enhances the development, growth, and photosynthesis under abiotic stress conditions, such as heat and drought in coffee plants [21]. $Ca^{+2}$ acts as a sensor of environmental signals, including the soil–water gradient affecting the processes responsible for plant water management. $K^+$ deficiency

in the root growth medium also triggers the $Ca^{+2}$ sensors, which in turn activate the high-affinity $K^+$ transporters in cells within the root cortex [19].

The water use efficiency (WUE) is a valuable index to compare the water productivity in different regions for the same crop, of different crops in the same region, and for other possible uses; integrate crop yield and water used to achieve that yield; provides a valuable index for optimizing water uses among different agricultural sectors in water-scarce regions [19,22], or climate change and variability scenarios. A proper understanding and knowledge of the crop management effects on WUE may provide researchers and farmers with opportunities to identify and select appropriate practices for improving in-field WUE [18]. According to Ritchie and Basso [23], increased crop yield in a particular environment is accompanied by a simultaneous increase in the WUE. This research aimed to evaluate the influence of the $K^+$ and $Ca^{+2}$ fertilization in coffee on the WUE in rain-fed conditions and its contribution as an incremental adaptation strategy to climate change and variability.

## 2. Materials and Methods

### 2.1. Trial Location

During five years from July 2014 to June 2019, two trials were carried out under field conditions in the southeast region of Colombia, in El Pital-Huila in a farm located at 02°20.1′62″ N–75°50.1′41″ W and 1700 m elevation, the meteorological data during the experiments were recorded (Table 1) and data available in https://www.cenicafe.org/es/index.php/nuestras_publicaciones/anuarios_meteorologicos accessed on 3 March 2021. The soil was a biotite–granite classified as a Typic Tropothents and Typic Dystrudepts [24], containing 70% sand, 24% silt and 6% clay, with a volumetric soil moisture at saturation level ($\theta_s$) = 0.69 $cm^3.cm^{-3}$; volumetric soil moisture at field capacity ($\theta_{FC}$) = 0.476 $cm^3.cm^{-3}$, and a soil humidity at wilting point of ($\theta_{wp}$) = 0.294 $cm^3.cm^{-3}$ at 40 cm depth.

**Table 1.** Climatic conditions obtained from the weather station. Simon Campos weather station 02°21′ N–75°53 W provided by the National Coffee Research Centre-Meteorological Network.

| Year | T. Min (°C) | T. Max (°C) | T. Med (°C) | R.H (%) | Rainfall (mm) | Sunshine (hours) |
|------|-------------|-------------|-------------|---------|---------------|------------------|
| **2014** | 15.7 | 23.6 | 19.1 | 74.6 | 1741.3 | 1233.1 |
| **2015** | 15.8 | 24.2 | 19.5 | 72.3 | 1319.6 | 1243.1 |
| **2016** | 16.1 | 24.1 | 19.6 | 73.4 | 1625.3 | 1241.4 |
| **2017** | 15.7 | 23.6 | 19.1 | 70.5 | 1976.3 | 1211.2 |
| **2018** | 15.7 | 23.5 | 19.0 | 75.3 | 1761.9 | |
| **2019** | | | | | 1482.1 | |
| **Mean** | **15.8** | **23.8** | **19.3** | **73.2** | **1651.1** | **1231.2** |

Two trials were installed in this location in two coffee varieties. Trial 1 was established in the *Coffea arabica L.* variety Castillo®with resistance to the coffee leaf rust (CLR) disease generated by the fungi *Hemileia vastatrix* Verkeley and Brome [25]. The plantation was established in 2012 without shade, and the coffee was planted with a plant density of 5100 plants $ha^{-1}$ at 1.4 m distance between plants and 1.4 m distance between rows. Trial 2 was established in *Coffea arabica L.* variety Caturra susceptible to CLR, without shade. The Caturra trial plantation was stem trimmed at 30 cm height in August of 2014 to initiate a new productive cycle, and this plantation was planted with a plant density of 6600 plants $ha^{-1}$ at 1.5 m distance between plants and 1.0 m distance between rows. Soil samples for soil fertility analysis before starting the treatment application were collected at 0–30 cm depth, the pH was determined in $CaCl_2$, organic matter by Walkley–Black, P by Bray-II, and the exchangeable fraction of K, Mg, and Ca with 1 N ammonium acetate extraction (1 N $NH_4C_2H_3O_2$, pH 7.0). The cations in the extracts were detected using an ICP (PerkinElmer, Optima 8300), soil texture analyses using the hygrometer–Bouyoucos method (Table 2).

**Table 2.** Soil fertility properties for the experimental sites.

| Trial | PH CaCl$_2$ | C.org | P | K | Ca | Mg | Al |
|---|---|---|---|---|---|---|---|
| | | % | | | mg.kg$^{-1}$ | | |
| Castillo | 4.0 (low) | 1.38 (low) | 1.8 (low) | 141 (med) | 448 (low) | 150 (high) | 196 (high) |
| Caturra | 4.4 (low) | 2.14 (low) | 5.0 (low) | 254 (high) | 274 (low) | 98 (med) | 244 (high) |

To evaluate the effect of the Ca$^{+2}$ and K$^+$ rates on yield, four Ca$^{+2}$ and three K$^+$ rates were evaluated in each of the trials: 150, 100, 50, 0 kg CaO ha$^{-1}$ year$^{-1}$ and 230, 180, 100 kg. K$_2$O.ha$^{-1}$ year$^{-1}$, respectively. The fertilizer sources used in the trial were ammonium nitrate-based NPK fertilizers of different grades and calcium nitrate (15 N-26% CaO) to supply the required K$^+$ and Ca$^{+2}$ rates. The other nutrients' average rates were 260, 110, 77, and 66 kg ha$^{-1}$ year$^{-1}$ of N, P$_2$O$_5$, MgO, and S, respectively.

Both experiments were set up in a randomized complete block design with four replications. For the Castillo variety, each plot in the blocks had 54.88 m$^2$ with 28 plants and 10 effective yield evaluation plants. For the Caturra variety, each plot in the blocks had 42 m$^2$ with 28 plants and 10 effective plants for yield evaluation. In the Castillo variety, the harvest data were collected monthly from January 2015 to July 2018 and for the Caturra variety from January 2016 to June 2019.

### 2.2. Water Use Efficiency Calculation

In agronomy, according to Viets [26], the water use efficiency (*WUE*) is generally defined as an index that is calculated as the yield of the main crop product (*Y$_a$*) per unit of water uses (*ET$_{act}$*):

$$WUE = \frac{Y_a}{ET_{act}} \tag{1}$$

Yield (*Y$_a$*, in kg ha$^{-1}$) is defined as the quantity of the actual harvestable crop part (in this study, coffee cherries) per given area during a fixed period. The actual seasonal crop water consumption-*ET$_{act}$*, in mm or m$^3$ ha$^{-1}$ [19,22,27].

In this study, the coffee cherry yield was taken during a monthly harvest from January to December from 2015 to 2018, with a higher proportion of the harvest from April to June. From a coffee physiology perspective, eight months pass from flowering to harvest [28], for this reason, the *ET$_{act}$* was calculated from May to December of the harvest year because the flowers that open in May correspond to harvestable coffee in January, the flowers from June result in the coffee harvest in February, and so on.

The *ET$_{act}$* was estimated following the FAO-56 approach [29] adjusted for coffee by Ramirez et al. [30] as follows:

$$ET_{act} = ET_o \times K_c \times \rho \tag{2}$$

where: *ET$_o$* is the reference evapotranspiration, *K$_c$* is the crop coefficient, and $\rho$ is the adjustment to the crop evapotranspiration as a function of the soil moisture and is calculated as follows:

$$\rho = \frac{\theta_{i-1}}{\theta_s} \tag{3}$$

where: $\theta_{i-1}$ is the volumetric moisture content of the day before, and $\theta_s$ is the volumetric soil moisture at saturation level. According to the FAO-56 approach, if $\rho <= 0.35$, the coffee crop is the underwater deficit, which reduces the actual crop evapotranspiration; if $\rho > 0.35$, there is no limitation to the crop evapotranspiration.

The volumetric soil moisture in Equation (3) was calculated daily using the water balance model as follows:

$$\theta_i = \theta_{i-1} - ET_{act,i} - R_i + Pe_i \tag{4}$$

where $\theta_i$ is the volumetric soil moisture for day I, $R_i$ is a runoff for day $i$, and $Pe_i$ is the effective rainfall for day $i$.

The equation of the water balance was estimated as follows:

$$\theta_o = \theta_{fc} - ET_{act,o} - R_o + Pe_o \tag{5}$$

where $\theta_0$ is the volumetric soil moisture for day zero, $\theta_{fc}$ is the volumetric soil moisture at field capacity, $R_o$ is runoff, and $Pe_o$ is the effective rainfall on day zero, respectively.

The runoff ($R$) and effective rainfall ($Pe$) were estimated based on the daily rainfall ($P$) on the models developed for coffee by Jaramillo y Cháves [31] and Ramirez and Jaramillo [32] as follows:

For runoff:

If the rainfall ($P$) > 6.0 mm

$$R = \frac{5.16}{1 + 16.25\, exp^{(-0.072 \times P)}} \tag{6}$$

If the rainfall ($P$) <= 6.0 mm; $R = 0$

For effective rainfall ($Pe$)

If the rainfall ($P$) > 6.0 mm

$$Pe = \frac{69.13}{1 + 12.45 exp^{(-0040 \times P)}} \tag{7}$$

If the rainfall ($P$) <= 6.0 mm; $Pe = 0$

The reference evapotranspiration ($ET_o$) was estimated using the Hargreaves model adjusted by Ramirez et al. [33] for the Colombian coffee region.

$$ETo = 0.0018(T_{mean} + 17.8)(T_{max} - T_{min})^{0.5} Ra \tag{8}$$

where $T$ is the air temperature in °C, and $Ra$ is astronomical solar radiation in mm day$^{-1}$ calculated following the model presented by Allen et al. [29].

The crop coefficient (Kc) for coffee was selected according to the plan density and sowing age proposed by Da Silva [34] described in Table 3.

**Table 3.** Crop coefficients were used for both coffee trials during the study (adapted from Da Silva, [34]).

| Variety | Year | Kc |
|---|---|---|
| *Coffea arabica* cv. Castillo | 2014 | 0.9 |
|  | 2015 | 0.9 |
|  | 2016 | 1.0 |
|  | 2017 | 1.1 |
|  | 2018 | 1.1 |
| *Coffea arabica* cv. Caturra | 2014 | 0.6 |
|  | 2015 | 0.7 |
|  | 2016 | 0.8 |
|  | 2017 | 1.0 |
|  | 2018 | 1.2 |
|  | 2019 | 1.2 |

*2.3. Statistical Analysis*

According to the experimental design, all data were submitted to their respective analysis of variance (ANOVA) test. Statistical analysis was conducted using the Statgraphics Centurion software package (Statgraphics Technologies, Inc.). The Shapiro–Wilk modified test was applied for normality and carried out using the heterogeneity of variances using

the residuals vs. prediction test for each variable. The Fisher's LSD test was used to detect the treatments that significantly affected the ANOVA.

## 3. Results

### 3.1. Soil Moisture and Water Balance Distribution

The soil moisture shows a monthly and yearly variation, mainly correlated with the region's rainfall patterns (Figure 1 and Table 4-data available in the sumplementary material sesion). In the study area, the rainfall' monthly variation can be explained by the intertropical convergence zone-ITCZ movement in the region. The ITCZ brings the rainy season from September to May and the drying season from June to August.

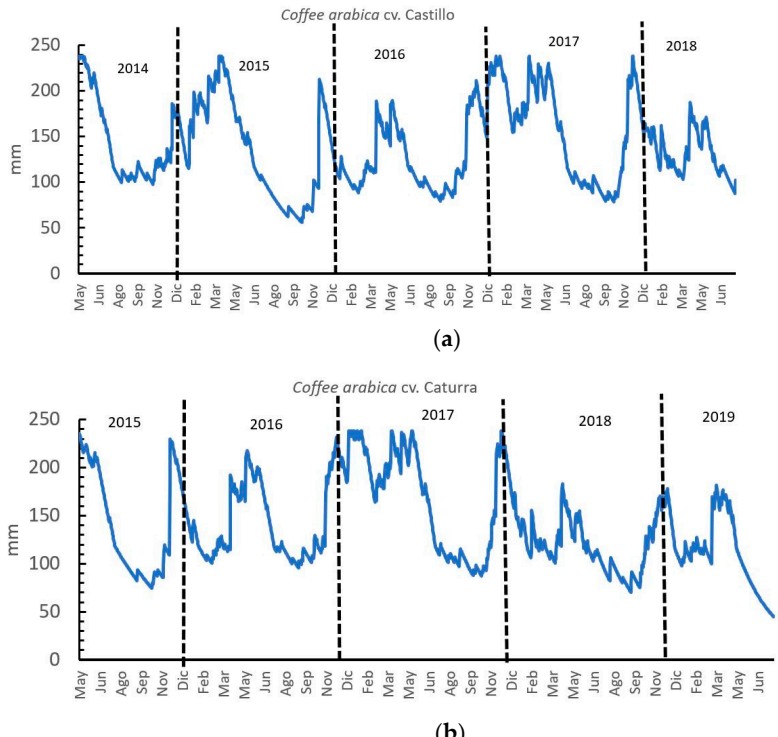

**Figure 1.** Soil moisture variation over five years for two coffee trials. *C. arabica* cv. Castillo (**a**), and *C. arabica* cv. Caturra (**b**).

**Table 4.** Water balance components for two coffee varieties during four harvest periods.

| Variety | Period | Harvest Year | $P$ | $P_e$ | $R$ | $ET_{act}$ | Deficit [+] |
|---|---|---|---|---|---|---|---|
| | | | mm | | | | |
| Castillo | May 2014–December 2015 | 2015 | 2.293 | 1.234 | 108 | 1.119 | 362 |
| | May 2015–December 2016 | 2016 | 2.234 | 1.191 | 105 | 1.058 | 408 |
| | May 2016–December 2017 | 2017 | 3.126 | 1.703 | 159 | 1.382 | 299 |
| | May 2017–July 2018 | 2018 | 2.053 | 1.117 | 102 | 1.067 | 393 |
| Caturra | May 2015–December 2016 | 2016 | 2.234 | 1.191 | 105 | 1.021 | 361 |
| | May 2016–December 2017 | 2017 | 3.126 | 1.703 | 159 | 1.310 | 296 |
| | May 2017–December 2018 | 2018 | 2.826 | 1.471 | 140 | 1.366 | 636 |
| | May 2018–July 2019 | 2019 | 1.806 | 888 | 89 | 988 | 438 |

[+] Water deficit was calculated using the $\rho$ index from Equation (3); when $\rho <= 0.4$ water deficit = $ET_{act}$ and when $\rho > 0.4$ water deficit = 0.0.

### 3.2. Influence of the Potassium and Calcium Nutrition on Water Use Efficiency

The coffee crop productivity is influenced by multiple variables like plant density, age of the plantation, nutrition, and climatic conditions, including water availability in the soil. In this paper, we focused on the effect of $Ca^{+2}$ and $K^+$ nutrition on WUE in two

coffee varieties. The $Ca^{+2}$ nutrition increases the WUE significantly for both varieties, reaching the highest WUE at 100 kg CaO ha$^{-1}$ year$^{-1}$ (Figure 2a,b, details please refer to supplementary). In both trials, the positive influence of $Ca^{+2}$ fertilization on the increase of the WUE in coffee is apparent. Several authors report that the coffee plantation responds to the $Ca^{+2}$ fertilization. Some authors even found a response at high $Ca^{+2}$ levels in the soil of 1000 to 2300 mg kg$^{-1}$ [35,36]. In the two trials analyzed in this paper, the $Ca^{+2}$ levels in the soils were low, which explains the significant increase in WUE due to applying soluble $Ca^{+2}$ fertilizers (Table 2).

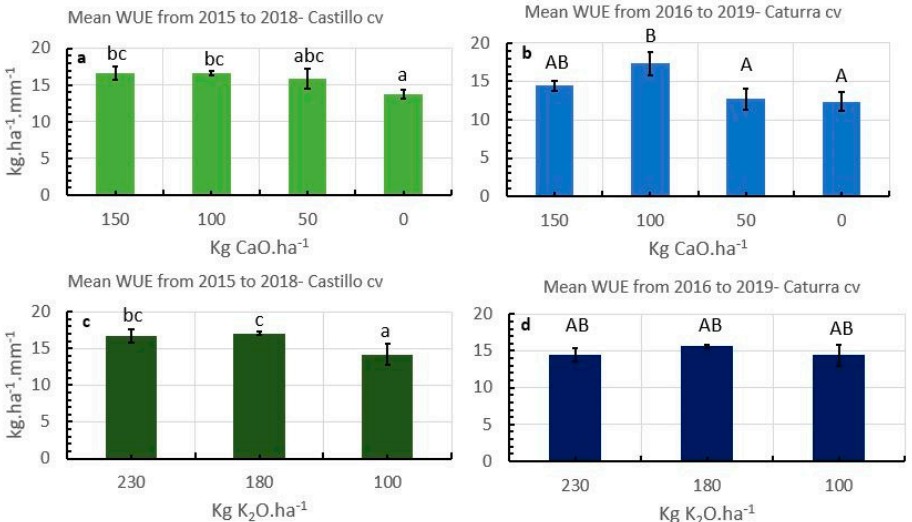

**Figure 2.** Influence of the calcium and potassium fertilization on water use efficiency in two coffee varieties, Castillo cv (**a,c**) and Caturra cv (**b,d**). Different letters indicate statistically significant differences in LSD test *p* < 0.05.

In the case of the $K^+$, statistical differences in the WUE were observed in the Castillo trial but not in the Caturra trial (Figure 2c,d, details please refer to supplementary). Higher WUE was achieved with $K^+$ rates between 230 to 180 kg of $K_2O$ ha$^{-1}$ year$^{-1}$. The $K^+$ rates in the Caturra trial had no statistically significant effect on the WUE because the $K^+$ level in the soil was higher than 160 mg kg$^{-1}$ (Table 2). At this $K^+$ soil content, the probability of response to the $K^+$ fertilization is lower than 5%. The recommended rate is 113 kg of $K_2O$ ha$^{-1}$ year$^{-1}$ [35].

The results indicate that the WUE in coffee is not constant and is affected by climate variability, yield curve, and disease incidence (Table 5). In 2016 the climate variability due to the positive phase of the El Niño–ENSO conditions increased the water deficit during the growing season up to 408 mm (Table 4); under these conditions, the WUE in coffee increases, compared to years with lower water deficit, such as 2015 and 2016. In the case of the Caturra variety, the WUE in 2016 was low (Table 5) because the first harvest took place in that year and usually in the first harvest year, coffee yields are low due to lower leaf area index and lower evapotranspiration rates, resulting in a rather low water deficit in the crop. The climate variability generated by the El Niño–ENSO conditions during 2015–2016 shows a positive influence on the Caturra variety yield in 2017, which in turn increased the WUE. This positive effect is mainly associated with an increased mean air temperature of 0.5 °C (Table 1), with a positive influence on vegetative growth that produced an excellent harvest during the third year after stem trimming, with a consequent increase in WUE (Table 5). After a very good harvest in 2017, yield and WUE in Caturra in 2018 and 2019 decreased, mainly associated with a substantial coffee leaf rust (CLR) incidence. CLR is a critical coffee disease generated by fungi (*Hemileia vastatrix* Verkeley and Brome). The disease incidence and severity developed in parallel to the coffee harvests [37]. Reductions in WUE were not observed in the Castillo variety because this variety is resistant to CLR.

**Table 5.** Yearly water use efficiency for both coffee varieties under variable $Ca^{+2}$ and $K^+$ rates.

| Variety | Harvest Year | kg CaO ha$^{-1}$ | | | | kg K$_2$O ha$^{-1}$ | | |
|---|---|---|---|---|---|---|---|---|
| | | 0 | 50 | 100 | 150 | 100 | 180 | 230 |
| | | WUE kg ha$^{-1}$ mm$^{-1}$ | | | | | | |
| Castillo | 2015 | 12.9 | 11.6 | 12.6 | 13.2 | 11.8 | 13.5 | 13.2 |
| | 2016 | 18.9 | 19.2 | 21.0 | 21.2 | 17.2 | 22.1 | 21.2 |
| | 2017 | 12.6 | 15.4 | 16.5 | 15.6 | 13.9 | 15.1 | 15.6 |
| | 2018 [+] | 9.2 a | 17.2 b | 16.3 c | 16.6 c | 13.6 A | 17.2 B | 16.6 B |
| Caturra | 2016 | 9.7 | 8.4 | 12.9 | 9.3 | 9.7 | 14.3 | 9.3 |
| | 2017 | 24.5 | 25.3 | 29.1 | 27.3 | 25.6 | 23.0 | 27.3 |
| | 2018 | 5.4 | 6.8 | 8.4 | 8.6 | 8.7 | 7.3 | 8.6 |
| | 2019 [+] | 9.7a | 10.1 b | 18.8 c | 12.4 b | 13.5 A | 17.6 B | 12.4 C |

[+] Different letters indicate statistically significant differences LSD test $p < 0.05$.

For both varieties, a long-term effect of $K^+$ and $Ca^{+2}$ fertilization was observed with a significant impact of both nutrient applications at the end of the trial period (2018 for Castillo and 2019 for Caturra), which together show a considerable effect when analyzing the average value of WUE during the production cycle (Figure 2), applying 100 kg of CaO ha$^{-1}$ year$^{-1}$ of increasing the WUE from 13.7 to 16.7 kg ha$^{-1}$ mm$^{-1}$ in Castillo cv. and from 12.4 to 17.3 kg ha$^{-1}$ mm$^{-1}$ in Caturra cv. 21% and 40% more WUE, respectively, and in the case of K, increase the rates from 100 to 180 kg K$_2$O ha$^{-1}$ year$^{-1}$ of increasing the WUE from 14.1 to 17.0 kg ha$^{-1}$ mm$^{-1}$ in Castillo cv. and from 14.4 to 15.6 kg ha$^{-1}$ mm$^{-1}$ in Caturra cv. 20% and 8% more WUE, respectively.

## 4. Discussion

The movement of the intertropical convergence zone (ITCZ) in the study region influences the seasonality of the rains with direct impacts on the coffee production cycle, especially on the pre-flowering stage in the dry season, flowering stage at the beginning of the rainy season, and cherry development during the rainy season [38,39]. However, in some years, the rainfall patterns change mainly associated with ENSO conditions. For the region where the research was carried out, it means that the positive phase of the ENSO known as "El Niño" reduces rainfall, increases solar radiation and air temperature. During the negative phase of the ENSO, known as "La Niña", the rainfall increases, and solar radiation and air temperature decrease [40–42].

These two climatic variability sources (ITCZ and ENSO) explain the variation in the soil moisture distribution during the five years of the study (Figure 1). For example, the soil moisture reduction between May 2015 to April 2016 was directly associated with the positive ENSO described by Ocean Niño Index (ONI) [43]. The ONI index describes the variations of the superficial Pacific sea temperature in region 3.4 [44]. The ONI index during this period showed positive values (>1.0 °C), indicating warmer conditions that resulted in a reduction in rainfall by 28.5% in the period May 2015 to April 2016 compared to the same period in 2016 to 2017. The reduction in rainfall increased the water deficit in the coffee plantations at 50 cm of root depth to 408 mm for the Castillo trial and 361 mm for the Caturra trial, compared with the same period 2016 to 2017, where the water deficit was 299 mm and 296 mm for the Castillo and Caturra trial, respectively. The increasing water deficits, in turn, resulted in a subsequent reduction of the crop evapotranspiration ET$_{act}$ in 2015 and 2016 (Table 4).

As a perennial crop with a lifespan of 20 to 30 years, coffee is subjected to the impact of climate change and climate variability [9]. In this work, a 49% variation in rainfall was observed between 2015, 2016, and 2017. This variation was caused by El Niño–ENSO conditions altering the amount and distribution of rainfall. A water deficit in coffee is required around flowering, mainly to break the dormancy of flower buds [38,45,46]. The coffee plantations can compensate between 15 and 30 days of water deficit, depending on soil type and crop water demand [47]. However, the coffee yield is strongly determined by climatic conditions, i.e., a water deficit during cherry and bean development or veg-

etative growth drastically reduces the productivity and quality of coffee [47–49]. These dependencies make the smallholder coffee farmers highly vulnerable, especially as an adaptation to climate variabilities in perennial crops like coffee may take several years [10]. This research shows that improved management practices, such as optimized nutrition solutions, including targeted $K^+$ and $Ca^{+2}$ nutrient applications, enhance the adaptation of coffee to climate impacts. Such incremental adaptation strategies [12] ensure sufficiently high productivity and farmer profitability under climate stress conditions. Adequate crop nutrition also helps to improve WUE (Figure 2). This has been shown explicitly in the trial with cv. Castillo in 2016, a year with increased water deficit due to El Niño–ENSO conditions. In this year, the treatments with 100 kg of CaO ha$^{-1}$ year$^{-1}$ and 100 to 180 kg $K_2O$. ha$^{-1}$ year$^{-1}$ resulted in a 12% and 28% increase in WUE, respectively, compared to the treatment without $Ca^{+2}$ or $K^+$ fertilization.

Ritchie and Basso [23] demonstrate that under most conditions, increases in yield due to improved fertilizer practices also result in increases in WUE. Studies on the influence of the $K^+$ and $Ca^{+2}$ fertilization on WUE in coffee at field conditions are rare, but Salamanca et al. [50] studied the impact of nitrogen supply on the water deficit of nursery coffee plants under greenhouse conditions and reported a significant reduction in the WUE in the treatment without N application. Grzebisz et al. [19] report in spring triticale and Maize increases in the WUE due to the $K^+$ fertilizer application and highlight the importance of $K^+$ fertilizer applications under rain-fed conditions, a measure to alleviate water-deficit stress at least partially.

From a socioeconomic perspective, understanding the extent of climate-driven impacts on coffee production and the benefits of potential adaptation strategies will be vital to maintain and improve coffee productivity and profitability and sustain the livelihoods of smallholder producers all over the world [9]. This research shows that coffee farmers are subject to reduced water availability and an increase of disease severity under the current climate variability scenarios, such as coffee leaf rust (CLR). CLR promotes substantial yield reductions in the years following the infection, especially in susceptible varieties like Caturra cv. However, this paper shows that implementing incremental adaptation strategies, such as optimized $K^+$ and $Ca^{+2}$ fertilizers applications, helps alleviate the disease incidence on the coffee crop in the longer term.

## 5. Conclusions

This study is a pioneer study conducted to understand the influence of specific nutrient management practices on yield and WUE in rain-fed coffee at the field level during a whole production cycle. It is shown that optimized nutrition practices in coffee help improve yield and farmer profitability under climate stress conditions and ensure efficient use of the increasingly scarce water resource. These results indicate that proper nutrient management, in this case, an adequate $K^+$ and $Ca^{+2}$ application, effectively contributes to sustaining the productivity and improve the WUE under climate variability conditions, disease incidences, like coffee leaf rust (CLR) and tree aging, and mainly contributing positively as an incremental strategy to climate variability and change adaptation.

**Supplementary Materials:** The following are available online at https://www.mdpi.com/article/10.3390/hydrology8020075/s1, Figure S1: Soil moisture variation over five years for two coffee trials. *C. arabica* cv. Castillo **a**, and *C. arabica* cv. Caturra **b**, Figure S2: Influence of the calcium and potassium fertilization on water use efficiency in two coffee varieties, Castillo cv (**a,c**) and Caturra cv (**b,d**). Different letters indicate statistically significant differences in the LSD test $p < 0.05$.

**Author Contributions:** V.H.R.-B. contributed to the field trial implementation, data acquisition, data analysis and manuscript preparation. J.K. contributes to the field trial setup, data analysis, and manuscript preparation. Both authors have read and agreed to the published version of the manuscript.

**Funding:** The authors declare that this research received funding from Yara International. The funder was not involved in the study design, collection, analysis, interpretation of the data, the writing of this article or the decision to submit it for publication.

**Conflicts of Interest:** The authors declare that the research was conducted in the absence of any commercial or financial relationship that could be constructed as a potential conflict of interest.

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
