# Peer review of "Calcium and Potassium Nutrition Increases the Water Use Efficiency in Coffee: A Promising Strategy to Adapt to Climate Change"

_hydrology, doi:10.3390/hydrology8020075_

Round 1

Reviewer 1 Report

This article provides a new idea for coping with global warming from the perspective of improving the water use efficiency of coffee, and it has important practical significance in helping small farmers cope with the reduction in coffee production caused by global warming. Before the article is published, the author should note the following points:

  1. In the introduction part, the author has spent a lot of space to introduce climate change and its influence on coffee cultivation, and the mechanism by which potassium and calcium increase water use efficiency is briefly introduced only in one paragraph. It is obviously not enough at all. The author needs to add relevant research literature and explain the research gap with other scholars. In fact, some important references were not cited by the authors, such as:

Waraich, Ejaz Ahmad, et al. "Improving agricultural water use efficiency by nutrient management in crop plants." Acta Agriculturae Scandinavica, Section B-Soil & Plant Science 61.4 (2011): 291-304.

Grzebisz, Witold, et al. "The effects of potassium fertilization on water‐use efficiency in crop plants." Journal of Plant Nutrition and Soil Science 176.3 (2013): 355-374.

  1. The citation of the literature should be in order

  1. line95: The author needs to use a map to accurately tell the reader the location of the research site.

4.line102: Climate data is open source, the author should provide the relevant website of the data source

  1. In the result section, the author should directly clarify the conclusion drawn from the experiment. This part does not need to add references, such as: line196, line201, line205, line206. The explanation and illustration of the result should be placed in the discussion section.

Author Response

  1. In the introduction part, the author has spent a lot of space to introduce climate change and its influence on coffee cultivation, and the mechanism by which potassium and calcium increase water use efficiency is briefly introduced only in one paragraph. It is obviously not enough at all. The author needs to add relevant research literature and explain the research gap with other scholars. In fact, some important references were not cited by the authors, such as:

1.  Both references were incorporated into the paper and the introduction was complemented according with the suggestions.

2. The Citation was arranged.

 3. We do not have a suitable software to make a map of the region, I´m Not sure if is really needed?

 4. The climate data were taking from the National Coffee Research Center-FNC in the following link that also was include in the paper in the Materia and Methods section. 

https://www.cenicafe.org/es/index.php/nuestras_publicaciones/anuarios_meteorologicos

5. The explanation and illustration of the results was placed into the discussion section.

Reviewer 2 Report

Identifying and implementing effective strategies for adapting crops to adverse climate change is a major research topic worldwide. It is noteworthy that the authors conducted this research over a relatively long period of five years, to investigate accurately the influence of crop fertilization in natural growing conditions.

The introductory part was properly designed, the relevance of the approached topic was emphasized and the paper's aim was clearly defined.

The research methodology could provide the key information required to understand the experimental study; The materials used and the growing conditions of the selected crops were described, also the calculation algorithms and statistical analysis method. I recommend that the authors remove the footnote from Table 2 (line 115) but complete this section with more detailed information on the methods and devices used to analyze the chemical parameters displayed in Table 2. Please, be careful to the correct writing of KCl solution 1 M.

Experimental results are structured and expressed clearly, the results are analyzed in-depth, being accompanied by suggestive graphical representations. There are some small typos, I recommend paying more attention to the spaces between letters or words (lines 236, 238, 239, 293, 296, 309, 321, etc.).

Discussions are elaborated coherently and logically, in accordance with the experimental results. The comparative analysis with some results obtained by other researchers in related studies was well integrated in the text.

The main conclusions of the study were pointed out. However, I would suggest highlighting them by including a separate section of the Conclusions. The selected and analyzed scientific literature is relevant, in close connection with the approached topic.

Author Response

1. The foot note was remove and better explanation of the soil nutrient extraction were added into the text.

Experimental results are structured and expressed clearly, the results are analyzed in-depth, being accompanied by suggestive graphical representations. There are some small typos, I recommend paying more attention to the spaces between letters or words (lines 236, 238, 239, 293, 296, 309, 321, etc.).

2. Adjusts were done. .

3. The conclusions session was include.